# Efficacy of the Newcastle Disease Virus Genotype VII.1.1-Matched Vaccines in Commercial Broilers

**DOI:** 10.3390/vaccines10010029

**Published:** 2021-12-27

**Authors:** Hesham A. Sultan, Wael K. Elfeil, Ahmed A. Nour, Laila Tantawy, Elsayed G. Kamel, Emad M. Eed, Ahmad El Askary, Shaimaa Talaat

**Affiliations:** 1Department of Birds and Rabbits Medicine, Faculty of Veterinary Medicine, University of Sadat City, Menoufiya 32958, Egypt; elsayedghram@gmail.com (E.G.K.); shimaa.talaat@vet.usc.edu.eg (S.T.); 2Avian and Rabbit Medicine Department, Faculty of Veterinary Medicine, Suez Canal University, Ismailia 51522, Egypt; 3Agriculture Research Center, National Laboratory for Veterinary Quality Control on Poultry Production, Animal Health Research Institute, Giza 12566, Egypt; drahmednour83@yahoo.com; 4Agriculture Research Center, Pathology Department, Animal Health Research Institute, Giza 12566, Egypt; rowaina1989@ahri.gov.eg; 5Department of Clinical Laboratory Sciences, College of Applied Medical Sciences, Taif University, P.O. Box 11099, Taif 21944, Saudi Arabia; e.eed@tu.edu.sa (E.M.E.); ahmedelaskary@azhar.edu.eg (A.E.A.)

**Keywords:** Newcastle disease virus, genotype-matched vaccine, inactivated vaccine, maternal derived immunity, commercial broiler

## Abstract

Class II genotype VII Newcastle disease viruses (NDV) are predominant in the Middle East and Asia despite intensive vaccination programs using conventional live and inactivated NDV vaccines. In this study, the protective efficacies of three commercial vaccine regimes involving genotype II NDV, recombinant genotype VII NDV-matched, and an autogenous velogenic NDV genotype VII vaccine were evaluated against challenge with velogenic NDV genotype VII (accession number MG029120). Three vaccination regimes were applied as follows: group-1 received inactivated genotype II, group-2 received inactivated recombinant genotype VII NDV-matched, and group-3 received velogenic inactivated autogenous NDV genotype VII vaccines given on day 7; for the live vaccine doses, each group received the same live genotype II vaccine. The birds in all of the groups were challenged with NDV genotype VII, which was applied on day 28. Protection by the three regimes was evaluated after infection based on mortality rate, clinical signs, gross lesions, virus shedding, seroconversion, and microscopic changes. The results showed that these three vaccination regimes partially protected commercial broilers (73%, 86%, 97%, respectively, vs. 8.6% in non-vaccinated challenged and 0% in non-vaccinated non-challenged birds) against mortality at 10 days post-challenge (dpc). Using inactivated vaccines significantly reduced the virus shedding at the level of the number of shedders and the amount of virus that was shed in all vaccinated groups (G1-3) compared to in the non-vaccinated group (G-4). In conclusion, using closely genotype-matched vaccines (NDV-GVII) provided higher protection than using vaccines that were not closely genotype-matched and non-genotype-matched. The vaccine seeds that were closely related to genotype VII.1.1 provided higher protection against challenge against this genotype since it circulates in the Middle East region. Updating vaccine seeds with recent and closely related isolates provides higher protection.

## 1. Introduction

Newcastle disease virus (NDV) is a member of *Avulavirinae* of the family *Paramyxoviridae* viruses of the genus Avian orthoavulavirus-1 (formerly designated as the Avian avulavirus-1, commonly known as Avian paramyxovirus) [1]. ND is caused by virulent strains of Newcastle disease viruses (NDV), which are enzootic in many countries and have been reported worldwide. Despite the introduction of vaccines for these viruses for control more than 60 years ago, ND remains one of the most important avian diseases and affects major poultry farms in various countries [2]. Phylogenetic analyses show that recent virulent isolates are more closely related to the virulent strains that were isolated during the 1940s and have been continuously used in laboratories and experimental challenge studies [3]. NDVs are single stranded, non-segmented, negative sense RNA viruses that encode at least six structural proteins [4]. The six encoded proteins are the nucleocapsid, phosphoprotein, matrix, fusion, hemagglutinin-neuraminidase, and the polymerase proteins. The fusion (F) and (HN) proteins form spike-like projections on the outer surface of the viral envelope, including the neutralizing and protective antigens of NDV [5]. Genetically, NDV has been classified into class I and class II viruses. Class I NDV are found in wild birds, tend to show low virulence, and are rarely detected in poultry species [6,7]. Based on the analysis of the nucleotide sequence of the F gene in class II viruses, they are divided into 18 genotypes (I–XVIII), with genotypes V, VII, and VIII being the most common genotypes circulating in the world [8]. The use of the updated classification criteria has significantly impacted the genotype VII landscape [1]. Many former sub-genotypes did not fulfill the distance (VIIb, VIId, VIIe, VIIj, VIIl), branch support (VIIh, VIIi, VIIk), and/or number of independent isolates (VIIk) criteria. These were merged into a total of three genotype VII sub-genotypes. The viruses that are responsible for the fourth NDV panzootic were grouped together, and based on nucleotide distance, they were classified into a single genotype (VII.1.1), combining the former sub-genotypes VIIb, VIId, VIIe, VIIj, and VIIl. An exception is the former sub-genotype VIIf, which was classified as a separate sub-genotype, namely VII.1.2. The groups of viruses that are involved in the fifth NDV panzootic (VIIh and VIIi) are those affecting Indonesia, Asia, the Middle East, Europe, and Africa [1].

Antigenic similarity is found among all NDV genotypes, and all viruses will cross-protect against challenge with each other. Thus, immunological stimulation can serve as a basis for vaccination with live low virulence NDV vaccines to protect against virulent NDV (vNDV) [5,9]. Yet, the antigenic and genetic diversity of the different genotypes has been observed in many countries with multiple cases of vaccination failure and the reduced ability of classical vaccines to decrease virus replication and shedding of the virulent NDV [10]. Early studies have shown antigenic differences between strains of NDV using virus neutralization assays and hemagglutination inhibition (HI) assays using monospecific antibodies and by evaluating sequences of neutralizing epitopes [11,12]. Recent Egyptian NDV isolates are related to genotype VII 1.1 according to the recent classification due to the presence of mutations at different sites of the F gene, such as in the N-glycosylation sites, epitope binding sites, and cysteine residues, that may affect virus pathogenicity and that may interfere with the protection offered by classical vaccines [1,13]. Besides the classical vaccines, newer NDV vaccines have been used in many countries. Through recombinant technology, these novel vaccines are based on recombinant herpesvirus-vectored and reverse genetic LaSota NDV vaccines that express velogenic F and/or HN genes [14]. Similar to most vaccines, NDV vaccines do not prevent vaccinated animals from becoming infected with a vNDV and from subsequently shedding the virus. However, most vaccines will significantly decrease the amount of virus that is shed in the feces and saliva compared to non-vaccinated birds [15]. Egypt already has two types of vaccines available from different genotypes (genotype II and recent genotype VII) that can be used to compare their level protection and virus shedding after challenge with VVND-GVII.1.1 [15,16]. Recently, recombinant GVII-matched chimeric vaccines, such as the Himmvac Dalguban N (Plus) Oil Vaccine (KBNP, Inc., Gyeonggi, Korea), have been developed by the reverse genetics method and have shown significant protection in specific pathogen-free chicken against the homologous genotype NDV [16]. Vaccines based on a recombinant LaSota strain backbone were also developed (KBNP-C4152R2L), where both antigenic genes, the F- and the hemagglutinin-neuraminidase (HN) genes, were replaced by other genes from the genotype VIId virus, KBNP-4152 [17,18]. However, before replacement, a mutation was induced to attenuate the recombinant strain at the F cleavage motif by changes from (112) RRQKR (116) to (112) GRQAR (116). To maintain this attenuation and to reduce the pathogenic instability, a single-point mutation was placed by insertion at codon 115. In addition, a 6-nucleotide sequence was inserted at the intergenic region between the matrix protein and F gene for attenuation without breaking the “rule of six.” The HN protein length was increased from 571 to 577 as a marker for later differentiation [17]. NDV belonging to genotype VIId was associated with severe problems in the oviducts, which lead to the production of soft shelled, shell-less eggs and even a decrease or loss in egg production, whereas the virus infection was associated with severe lymphocyte infiltration [14]. In broilers chicks, NDV associated with severe economic losses resulted from mortalities, severe enteritis, and subsequent bacterial activities (E.coli, Clostridium, Mycoplasma, Pasteurella) in the GIT [19,20,21,22] and in the respiratory organs in addition to losses from mixed infection with other viral diseases such as infectious bursal disease (IBD), avian influenza virus (AIV), and infectious bronchitis viruses (IBV) [21,23,24,25,26]. In addition to follicular degeneration effects in ovaries and oviducts, the qualities of the egg shell and egg albumen in velogenic Newcastle disease virus (VNDV)-infected birds were low due to the inadequate production of the steroid hormones and yolk resorption [27]. Previous data obtained from the current lab described that the use of genotype-matched NDV inactivated vaccines provided significantly higher protection and a smaller egg production drop in commercial layer hens compared to in non-genotype-matched vaccines (genotype II) after challenge with the ND-GVII virus [28]. This study aimed to evaluate the efficacy of the inactivated GII (LaSota) and inactivated recombinant reverse genetic ND vaccine on LaSota backbone-carrying F-HN genes from an Asian NDV-GVII isolate and an autogenous VNDV genotype VII.I.I against challenge with VNDV clade VII.1.1 in commercial broiler chickens with maternally derived antibodies (MDA) under laboratory conditions.

## 2. Materials and Methods

### 2.1. Ethical Statement

Animal studies were approved by the Animal Welfare and Research Ethics Committee of the University of Sadat City (approval ID: 20120632), and all procedures were conducted strictly in accordance with the Guide for the Care and Use of Laboratory Animals. Every effort was made to minimize animal suffering.

### 2.2. Challenge Virus and Vaccines

A previously identified and characterized NDV GVII virus, shown in Figure 1 (NDV Chicken/USC/Egypt/El Gharbia/2015), accession number MG029120, was used as a challenge virus in this study, and the genetic structure of the selected virus was the same as the one in prepared autogenous vaccine [28]. Three different commercial inactivated NDV vaccines were used for broiler vaccination in the study along with the same ND live vaccine. They included a GII NDV-inactivated oil emulsion vaccine (LaSota strain/NDV genotype II, EID_50_:8.2 log_10;_ Volvac B.E.S.T AI ND, Boehringer Ingelheim, Germany), which was kindly supplied by the local agency in Egypt (International Free Trade Company (IFT), Cairo, Egypt); an inactivated oil emulsion recombinant GVII NDV vaccine that uses the reverse genetic reassortant vaccine where the virus has been attenuated by the deletion of the multi-basic cleavage sites with the F gene then inserted on the LaSota backbone as previously described [29] (Himmvac Oil Vaccine/NDV genotype VIId “KBNP-C4152R2L strain with a concentration of 8.2 log_10_ EID_50_, KBNP Inc., Gyeonggi, Korea), which was kindly supplied by the local agency in Egypt (Egyptian Company for Trade and Distribution (Egy-Co), Cairo, Egypt); and an autogenous genotype VII 1.1 NDV inactivated oil emulsion vaccine (NDV-B7-Egy-2012 KM288609, strain with a concentration of 8.2 log_10_ EID_50_). This final vaccine was prepared as a whole inactivated velogenic genotype VII 1.1 vaccine as previously described in the OIE manual (2012), where the antigen that was propagated in SPF embryonated chicken eggs (ECE). It was then harvested from the allantoic fluid and inactivated and mixed with Montanide ISA-70 oil (Seppic Corp., Garenne-Colombes, France) adjuvant following the manufacturers’ instructions. This vaccine is considered to be an experimental autogenous inactivated vaccine [30].

### 2.3. Experimental Design

One hundred seventy-five-day-old commercial broiler chicks with maternal NDV antibodies were supplied from a local hatchery, from a breeder flock that had been vaccinated with inactivated NDV genotype II vaccines four times (on days 14, 55, 85, and 120 of life) and six times with live ND genotype II vaccines (on days 7, 35, 65, 95, 115, and 140 of life). They were housed in separated pens with food and water provided ad libitum at a biosecured experimental facility under the same conditions. The chicks were randomly divided into 5 groups (G1 to G5) of 35 birds each, as shown in Fig-1. A commonly used vaccination regime in commercial broilers in Egypt was applied in this study, which is based on two doses from a live ND vaccine (first on day 7 and booster on day 17 via eyedrop route) and one dose of an inactivated ND vaccine. Of these, three groups (G1, G2, and G3) were vaccinated 2 with the same live GII NDV vaccine (LaSota, Boehringer Ingelheim, Germany) twice on days 7 and 17 via the oculonasal route, and then with a single-dose of either an inactivated GII NDV vaccine (for G1 birds) or recombinant GVII NDV vaccine (for G2 birds) and an inactivated GVII NDV vaccine (for G-3 birds) subcutaneously on day 7. The other group (G4) of birds received no vaccination during the experiment time, and another group kept as a non-vaccinated non-challenged group (G5). The four groups were challenged with the virulent NDV, USC2015 strain “Accession No.: MG029120”, (10^6^ EID_50_ per bird) via the intranasal route (to simulate the natural route of infection). Meanwhile, cloacal and tracheal swaps were collected from all groups (10 birds per group) before the challenge, which were checked by real-time reverse transcriptase polymerase chain reaction (RT-PCR) against common NDV using primer sets [31], as shown in Table 1. All of the experimental groups were monitored and recorded for clinical signs for 10 days after NDV challenge. Tracheal and cloacal swabs were collected at 2, 4, and 7 dpc to monitor virus shedding (as 10 birds per group).

### 2.4. Serology

Newcastle disease serum antibodies were quantified by hemagglutination inhibition (HI) assay in U-bottomed 96-well microtiter plates as per the procedure recommended in the OIE Manual [30]. Blood samples were collected on days 7, 14, 21, 28, 35, and 42 of life, and these samples were serum separated and stored at −20 °C until analysis. The tests were performed using four hemagglutination units (4HAU) of the NDV LaSota antigen and NDV-genotype VII antigen. The end point was determined as the reciprocal of the highest serum dilution exhibiting the complete inhibition of hemagglutination. Hemagglutination inhibition titers were expressed in terms of log_2_, and samples showing HI titers that were equal/over 4-log_2_ were considered positive [30].

### 2.5. Histopathological Examination

Euthanasia, when needed, was performed by cervical dislocation, as previous described [32]. Tissue samples (trachea, cecal tonsils, brain, thymus, and spleen) were fixed in 10% buffered formalin for about 15 to 20 days, processed for histology by routine procedures, and stained with hematoxylin and eosin (H&E) [33]. An ordinal scoring system for the lesions of infected tissues was obtained according to the progression of severity; lesions were scored as 1 = negative, 2 = mild, 3 = moderate, 4 = severe, and 5 = very severe [34]. Five random optical fields were examined and scored, and then the mean of the five fields was calculated. The formula of the mean severity index (MSI) was given by the function sum of the mean lesion scores of the three examined organs of three chickens per group at 4 and 7 dpc divided by the total number of examined organs, as previously described [35].

### 2.6. Virus Shedding

Tracheal and cloacal swabs collected from 10 birds/group at 2, 4 and 7 dpc were immersed in 400 µL of Dulbecco’s modified Eagle medium with an antibiotic solution. Viral RNA was extracted directly using the TRIzol reagent (Gibco, Invitrogen, Carlsbad, CA, USA), as per the manufacturer’s instructions, and were suspended in diethyl pyrocarbonate (DEPC) water. Standard RT-PCR was performed using a One-Step RT-PCR kit (QIAGEN, Valencia, CA, USA), with specific primers and probe sets as shown in Table 1. In brief, the RT-PCR assay was performed in a final volume of 25 µL containing 12.5 µL of the QuantiTect RT-PCR Master Mix, 0.25 µL of the QuantiTect RT Mix, 0.25 µL of each primer (50 pmol concentration), 0.125 mL of each probe (30 pmol concentration), 3.625 µL of PCR-grade water, and 7 µL of RNA template. Reverse transcription reactions were set up at 50 °C for 30 min, followed by a primary denaturation step at 94 °C for 15 min, then 40 cycles of denaturation at 94 °C for 15 s, annealing at 54 °C for 30 s, and extension at 72 °C for 10 s. The reaction was carried out using an MX3005P real-time PCR machine (Stratagene, La Jolla, CA, USA). Real time reverse transcription (RRT-PCR) titers were converted into log_10_ EID_50_/mL, as previously described [36]. In brief, a triplicate of 10-fold dilutions of challenge NDV (10^6^ EID50/mL) were used to generate a standard curve using stock virus dilutions from 10-1 to 10^−6^. Because Ct is defined as the point at which the curve crosses the horizontal threshold line, virus log_10_ titers of a specimen were plotted against the Ct value, and the best-fit line was constructed. The linear range of the assay ranged from 1 to 10^6^ EID_50_/mL, with a correlation coefficient of 0.99. The system detection limit was 0.5 EID_50_/mL, which has previously been described as the standard [28]. The quantity of Newcastle disease virus in unknown samples was derived by plotting the Ct of unknown samples against the standard curve and was expressed in terms of log_10_ EID_50_/mL equivalents.

### 2.7. Statistical Analysis

Whenever necessary, data were analyzed using Student’s t-test or ANOVA, followed by the application of Duncan’s new multiple range test to determine the significance of differences between individual treatments and the corresponding controls [37].

## 3. Results

### 3.1. Clinical Signs, Postmortem Gross Lesions, and Mortalities

In the G-1 (vaccinated with inactivated GII and challenged), 29% (10/35) of the birds showed mild or moderate respiratory signs (nasal discharge, respiratory rales), and 71.5% (25/35) showed greenish diarrhea, but no eye lesions (lacrimation, slight occlusion to eye). Most of the birds recovered in a short period of time, demonstrating 73% (26/35) clinical protection within 10 dpc. Only 5.7% (2/35) showed nervous signs (paresis) after the recovery stage, and mortality was 23% (8/35) 10 dpc. (Table 2). In the G-2 (vaccinated with rNDV inactivated G VII and challenged), 28% (10/35) showed mild respiratory signs and nasal discharge, and 57% (20/35) showed greenish diarrhea, while 5.7% (2/35) showed eye lesions (lacrimation, slight occlusion to eye). Most of the affected birds recovered, indicating 86% (30/35) clinical protection 10 dpc. The mortality rate was 14% (8/35) 10 dpc (Table 2). In G-3 (vaccinated with autogenous inactivated NDV g VII and challenged), 8% (3/35) of the birds showed mild respiratory signs, and 5.7% (2/35) of them showed nasal discharge. Additionally, 28% (10/35) of the birds showed greenish diarrhea but no eye lesions, and most recovered quickly, indicating 97% (34/35) clinical protection within 10 dpc and only 3% (1/35) mortality 10 dpc (Table 2). In G-4 (non-vaccinated and challenged), 100% (35/35) of the birds showed moderate to severe respiratory signs, and 22.9% (18/35) showed nasal discharge, while 91.4% (32/35) showed greenish diarrhea, 8% (3/35) showed nervous signs and paresis, and most of the birds died (91.4%, 32/35), with 8.6% (3/35) recovering. The birds in G5 did not show any clinical illness or mortalities, ensuring the elimination effect of any other variables other than the vaccines and challenge virus, as shown in (Table 2).

The dead birds in groups G-1, G-2, and G-3 exhibited no or mild petechial hemorrhages in the proventriculus vs. severe petechial hemorrhages in G-4 and no lesions in the examined birds from G-5. Groups G-1, G-2, and G-3 exhibited mild, moderate, and moderate catarrhal tracheitis and lung congestion, respectively, vs. the severe catarrhal tracheitis seen in G-4, in addition to mild, moderate, and severe focal necrosis in the spleens of the birds from G-1, G-2, and G-3 vs. the very severe necrosis seen in the birds from G-4 and the almost no lesions seen in the birds in G5, respectively. The necrosis confirmed and was different from the apoptosis that was examined by the slide field by light microscope, and there were no apoptotic bodies present, which usually appear as vesicles containing necrotic cell debris, thus confirming that the lesions were necrosis and not apoptotic lesions. Mild cecal ulcers and no lesions were seen in the G-1, G-2, and G-3 birds vs. the moderate cecal ulcers that were seen in the G-4 birds and the no lesions seen in the in G5 birds, respectively, as shown in Table 3.

### 3.2. Serum Antibody Response to Vaccination

The dynamics of the NDV serological response were examined for 6 week. pc using the HI test with blood samples that had been collected from all groups at weekly intervals starting from 7 days of life until 42 days of life. The results showed non-significant differences between the different vaccinated groups, regardless of the genotypes of the inactivated NDV vaccines. HI titers were 4.2 in G-1 and G-2 but 4.8 in G-3 log_2_, as shown in Table 2. Before and after the NDV challenge, the mean HI titers of the groups (vaccine 1: LaSota virus, vaccine 2: rec-LaSota, vaccine 3: autogenous VII.I.I prepared vaccine and G-4 non-vaccinated group) and in G5 as a non-vaccinated non-challenged group are shown in Figure 2a,b and in Table 4.

### 3.3. Virus Shedding

All of the experimental birds were examined by RT-PCR for virus shedding using tracheal and cloacal swab samples collected at 2, 4, and 7 dpc. The results are summarized in (Table 4) and (Figure 3 and Figure 4). Before the NDV challenge, all of the tested birds were negative for NDV, avian influenza virus (AIV), and infectious bronchitis virus (IBV), the pathogens that are associated with mild to moderated respiratory signs in the vaccinated groups and severe respiratory signs in the non-vaccinated group.

On the 2nd dpc, the birds in G-1 showed that 40% were shedders (4/10) from both the tracheal and cloacal routes, with titers of 6.3 × 10^5^ ± 1.2 EID_50_ and 4 × 10^5^ ±1.4 EID_50_, respectively. The birds in G-2/3 showed that only 20% were shedders (2/10) from both the tracheal (2.3 × 10^5^ ± 0.0 EID_50,_ 2.1 × 10^5^ ± 0.0 EID_50_) and cloacal routes (5.5 × 10^4^ ± 0.0 EID_50,_ 4.5 × 10^4^ ± 0.0 EID_50_). The birds in G-4 showed that 60% were shedders (6/10), with 1.5 × 10^6^ ± 1.4 EID_50_ in the tracheal route and 1.3 × 10^6^ ± 0.8 EID_50_ in the cloacal route, while the birds in G5 showed non-detectable shedding from both routes, as shown in Table 4 and in Figure 3 and Figure 4.

On the 4th dpc, the birds in G-1 showed that 40% were shedders from both the tracheal and cloacal routes at 9.4 × 10^4^ ± 2.3 EID_50_ and 1.6 × 10^4^ + 1.32 EID_50_, respectively. The birds in G-2 showed that 20% were shedders (2/10) from the tracheal route at 1.4 × 10^4^ ± 0.0 EID_50_ and 40% from the cloacal route (4/10) at 4 × 10^4^ ± 1.3 EID_50_, while the birds in G-3 showed that 20% were shedders from both the tracheal and cloacal routes at 2.1 × 10^5^ ± 0.0 EID_50_ and 4.5 × 10^4^ ± 0.0 EID_50,_ respectively. The birds in G-4 showed that 100% were shedders from both tracheal and cloacal routes at 6.2 × 10^6^ ± 1.76 EID_50_ and 4.6 × 10^6^ ± 1.31 EID_50,_ respectively, as shown in Table 4 and in Figure 3 and Figure 4. On the 7th dpc, the birds in G1/2/3/5 showed undetectable levels of virus shedding, while the birds in G-4 showed 100% shedding from both the tracheal and cloacal routes at 1.7 × 10^7^ ± 0.87 EID_50_ and 2.6 × 10^6^ ± 0.63 EID_50_, respectively_,_ as shown in Table 4 and in Figure 3 and Figure 4.

### 3.4. Histopathological Examination

Lesions were observed in non-vaccinated commercial broilers challenged on 28 days in all of the examined organs (spleen, trachea, cecal tonsil, cerebrum, Bursa).

To evaluate the microscopic changes in broilers chickens who had bene challenged with VVNDV, samples were collected at 4 and 7 dpc from various organs, including the trachea, lung, bursa, liver, and spleen, where the main lesion scores (MLS) were 2.6, 2.1, 2.0, 3.2, and 1 on the 4th dpc in G1-5, respectively, and were 2.0, 1.9, 1.5, 2.8, and 1 on the 7th dpc in G1-5, respectively, as shown in Table 3 and Figure 5a,b. Microscopic examinations revealed very severe histopathological lesion scores in non-vaccinated broilers who had been challenged with VVNDV in all of the examined organs. The score was significantly higher than those of G1-3 (*p* < 0.05). Moderate histopathological lesions were observed in the spleens, trachea, cecal tonsil, cerebrum, and bursa of broilers who had been vaccinated with inactivated genotype II vaccines (G-1) compared to the lower lesions score in G-4 (*p* < 0.05) but were significantly higher than G2/3 (*p* < 0.05). Only mild histopathological lesion scores in the spleen, trachea, cecal tonsil, cerebrum, and bursa were observed in the groups who had been vaccinated with the genotype VII vaccine (G2/3), with the scores being significantly lower than those seen in G-1 and G4 (*p* < 0.05).

## 4. Discussion 

In the Middle East, Africa, and Asia, NDV-GVII has been associated with the vast majority of recently reported outbreaks [38,39]. However, the countries within those regions share common practices, including the use of commercial vaccines (prepared from GII, either live or inactivated) and similar vaccination regimes (number of doses, route of administration, combination of live and inactivated options). Regardless of vaccination regimes that depend on conventional, old non-genotype-matched vaccines for the circulating NDV in those regions, frequent outbreaks of ND or non-optimum control strategies occur, with considerable virus shedding and severe economic losses (mortalities and/or drop in egg production) [40].

In this study, protection against the development of clinical manifestations (Table 2), in the vaccinated challenged groups (G1-3) was significantly higher than in non-vaccinated challenged birds (G-4), while the birds in G5 showed no lesions or mortalities when the proper experimental conditions were ensured and any additional variables that could potentially affect the results were eliminated. Accordingly, the birds in G-4 showed almost 100% respiratory manifestations, 71% showed digestive disorders, and only 3% of the group recovered, showing nervous manifestations later on and a 91.4% mortality rate, reflecting the velogenic characteristics of the challenge virus and its pneumotropic, viscerotropic and neurotropic affinities in broiler non-vaccinated chicks [20,41]. The birds in G-1 (genotype II inactivated vaccine and challenged) showed the highest respiratory manifestations along with the highest levels of digestive disorders (29% and 71%, respectively) and the lowest rate of recovery (7–8%), in agreement with previous reports by Fawzy et al. (2020). The authors reported significantly lower protection when using G-II-inactivated ND vaccines in broilers flocks compared to in G-VII-inactivated ND vaccine after challenge with NDV-VII. However, the birds in G-2 (recombinant G-VII and Challenged) showed lower respiratory and digestive manifestations at rates of 28% and 57%, respectively, in comparison to the birds in G-1. The birds in G-3 (received autogenous NDV-GVII) showed the lowest rates of respiratory and digestive manifestations at 8% and 28%, respectively; those data clarified that increasing homology and a close matching degree between the challenge NDV and the vaccine seed improve clinical protection against developing clinical manifestation [41]. This agrees with the previous report of Miller et al. (2007), who used the NDV-GV virus and reported significantly higher clinical protection with the degree of virus and vaccine matching and thus may be associated with the mutations in the F-gene of the current circulating virus, something that Moharam et al. (2020) described in their results, which described that the NDV-VII.1.1 circulating in Egypt showed amino acid substitution at (K78R) and D170N in the F- gene neutralizing sites when using monoclonal antibodies that were prepared against similar isolates to those that originated from our lab, he found that six out of nine of the selected MAbs were able to block receptor binding, as demonstrated by HI activity. One MAb recognized an epitope that was only present in the homologue virus, while four other MAbs showed weak reactivity to selected other genotypes. On the other hand, one broadly cross-reacting MAb reacted with all of the tested genotypes and resembled the reactivity profile of different genotype-specific polyclonal antibody preparations, pointing to minor antigenic differences between tested NDV genotypes [16,42].

Developing nervous manifestations following recovery from respiratory and digestive illness has been always associated with velogenic NDV in non-vaccinated broilers or in cases of improperly vaccinated infected flocks [43,44]. The data that were obtained from this study highlight that birds who were vaccinated with a G-II inactivated vaccine (G-1) showed the highest level of nervous manifestations (8%), while the birds who were vaccinated with the either the genotype VII rNDV or Autogenous NDV (G2 and G3) did not show any nervous system manifestations, suggesting a high level of protection in the G-2 and G-3 birds. These data agree with the previous reports by Fawzy et al. (2020), who reported the minimal development of nervous manifestations in birds who had been vaccinated with homologous an autogenous genotype VII ND vaccine.

Regarding gross lesions, in G-4, the dead birds showed typical gross lesions, including petechial hemorrhages in the proventriculus, congested tracheitis, severe pneumonia, and ulceration in the cecal tonsils. Meanwhile, milder gross lesions were observed in the vaccinated groups (G1, G2 and G3). The degree of matching between the applied vaccines and the challenge virus decreased from the autogenous NDV vaccines to the rNDV-GVII vaccines and then to the G-II vaccines, and thus may be associated with the variation in the severity of the gross lesions, as the birds in G-1 showed more severe gross lesions, followed by G2 then G3, which confirmed the proportional relationship between the close degree of matching (between vaccine and challenge virus) and level of protection. Similarly, Sedeik et al. (2018) showed significantly higher protection levels against the clinical signs of ND c, the severity of the postmortem lesions, and the rate of mortality in the groups who had been vaccinated with a genotype-matched NDV vaccine (NDV GVII-origin vaccine) than the groups who had been vaccinated with a non–genotype-matched inactivated NDV GII-origin vaccine after challenge with velogenic NDV GVII from Egypt [41]. Miller et al. (2012) reported that the application of the non-genotyping vaccine required higher levels of humoral immune responses to produce protection against the challenge with NDV [20,45,46].

Consequently, the serological monitoring of the birds before experimental infection emphasized that the HI titers in the vaccinated groups using combined multiple doses from live and inactivated NDV vaccines can mount higher humoral immune responses (HI GMT 4.2 in G-1 and G-2 but 4.8 in G-3 log_2_, as shown in Figure 2). The seroconversion during the period between 14–21 days of life showed stagnant levels; however, the chicks were given a live booster vaccine on day 17 of life, and due to the live vaccine, the cell-mediated immunity was higher than the humoral immunity. In addition, the chicks who had been hatched with MDA benefited from the humoral immunity from the vaccines, which compensated for the waning in the MDA at the 3rd week of life [20]. The seroconversion results and correlation with protection against mortalities emphasize the hypothesis of herd immunity and agrees with the previous reports of van Boven et al. (2008), who found that herd immunity exists when at least 75–80% of the flock has developed specific HI titers that are equal to or higher than 4 log_2_ to NDV (when using four hemagglutination units per 50 mL of the antigen) [47]. There were no significant differences that were observed between the different vaccinated groups (G-1, G-2, and G-3) at the level of detectable humoral immune response (based on HI assay), but variations were observed in the protection levels against the development of clinical manifestations and protection against mortalities (78%, 86%, and 97% in G1, G2, and G3, respectively). The results of the seroconversion using the NDV-genotype VII showed higher levels in the group that had been vaccinated with the genotype VII vaccine (G-2 and G-3) and thus agreed with previous reports by Nagy et al. (2020), who recorded variation in the cross HI assay among the LaSota and NDV genotype VII antigen that has been circulating in Egypt in recent years [47]. This confirmed the previous reports of Sultan et al. (2020), who reported similar levels of HI titers that were associated with different genotypes of inactivated ND vaccines but that were not associated with similar protection levels. This also magnifies the value of the highly specific humoral immune response that is necessary to control the ND virus [28] and agrees with Miller et al. (2007), who claimed that overcoming the variation in the genotypes of ND vaccines that are used and in the circulating viruses (that do not match vaccines) necessitates higher HI titers being obtained from several vaccine doses of the non–genotype-matched vaccine, while genotype-matched vaccine required a lower amount of titers to provide protection [16]; these differences in protection against mortalities may be associated with the variation in the F-gene epitope sites between the current circulating NDV virus and the old vaccines [42]

The pathogenesis of VVNDV G-VII.1.1 in broiler chickens reflects the protection levels of vaccinated birds and shows the microscopic changes (Figure 5a,b) in different internal organs. The histopathological examination of the challenged non-vaccinated birds (G-4) revealed different pathological pictures in the examined organs, where the spleen showed severe congested blood vessels and lymphocyte depletion. The trachea showed severe edema and severe congestion in the lamina propria and sloughing of the tracheal mucosa. The cecal tonsil showed lymphocyte depletion and necrosis. The cerebrum demonstrated a perivascular cuff, large and small blood vessels were surrounded by inflammatory cells, and the bursa exhibited lymphocyte depletion in the medulla of the follicles, congested blood vessels with microcyst formation, and interfollicular connective tissue proliferation [47]. The birds in G-1 who had been vaccinated with genotype II inactivated vaccines showed significantly lower lesion scores (2.6 ± 0.78) in comparison to the birds who had been vaccinated with both inactivated genotype VII vaccines (G2-G3),who revealed lesion scores of 2.0 ± 0.23 and 1.7 ± 0.12, respectively; this may explain the lower mortality rates that were observed in G-2 and G-3 compared to those seen in the birds in in G-1. This finding emphasizes the role of genotype-matched vaccines in the protection against VVNDV challenge, as genotype-matched vaccines provide significantly higher protection to the internal organs and reduce the severity of infection and economic losses, a finding that is in agreement with previous reports [28].

Interest in the amount of vNDV that is shed into the environment by vaccinated birds has arisen as a potential indicator of vaccine efficacy. Previous experimental works have shown that by using the same (homologous) antigen in the vaccine seed that is similar to the genotype of a genotype II or genotype V NDV virus, it is possible to decrease not only the number of shedders, but also the amount of viruses that are shed from individual birds, which was from oropharyngeal and cloacal swabs [10].

In this study, the shedding results that were determined by RRT-PCR showed (Table 4 and Figure 3 and Figure 4) that the positive control group (G-4) demonstrated virus shedding (100% shedders) at 2, 4, and 7 dpc, which confirms the velogenic behavior of the challenge virus. Using inactivated vaccines significantly reduced virus shedding in terms of the number of shedders and the amount of virus shedding in all of the vaccinated groups (G1-3) compared to in the non-vaccinated group (G-4). Birds who had been vaccinated with the genotype II inactivated vaccine showed a significant reduction in virus shedding in terms of the number of shedders and the amount virus that was shed, compared to G-4, which is in agreement with previous reports highlighting the value of inactivated ND vaccines in reducing virus shedding [41]. Birds who had been vaccinated with genotype-matched vaccine (G-2/3) showed a significant reduction in virus shedding in terms of both the number of shedders and the amount of virus shedding at 2 and 4 dpc in comparison to the birds who had been vaccinated with the inactivated genotype II ND vaccine, but on 7 dpc, shedding was undetectable, a finding that is in agreement with previous reports by Fawzy et al. (2020) on the value of using genotype-matching inactivated ND vaccines to significantly reduce virus shedding (amount of virus shedding, No. of shedders and shedding period) as well as the virus load found on farms and in the environment. They also found that the use of genotype-matched inactivated ND vaccines would improve the control strategies for VVNDV on the poultry farms [41]. This may be due to the combined vaccine formulations, which have properties that all them to induce high levels of polyspecific antibody titers than genotype II vaccines, which could not efficiently neutralize the GVII challenge virus. It seems that the monospecific genotype VII antibodies can efficiently neutralize the virus and limit the transmission of infection to contact birds. This is in accordance with Miller et al. (2013), who suggested that high levels of polyspecific antibodies can prevent mortality but that the presence of monospecific antibodies is also necessary to decrease viral replication [6]. This approach will increase interest in developing monospecific vaccines to the virulent NDV (vNDV) genotype that is currently circulating in the field.

## 5. Conclusions

The use of genotype-matched inactivated NDV vaccines (recombinant GVII or autogenous GVII) instead of genotype-mismatched (GII) NDV vaccines induces higher clinical protection and reduces the severity of microscopic changes, leading to a significant reduction in NDV shedding in commercial broilers who have been exposed to velogenic NDV-GVII infection. Additionally, the use of a closely matched NDV vaccine is able to cover the recent mutation that has been observed in the F-gene and may be associated with elevated survival rates in comparison. In addition, there is a need to evaluate the efficacy of the commercial vaccines that are currently against challenge against the newly emerging GVII in the field in Egypt to ensure proper control strategies in broiler chickens.

## Figures and Tables

**Figure 1 vaccines-10-00029-f001:**
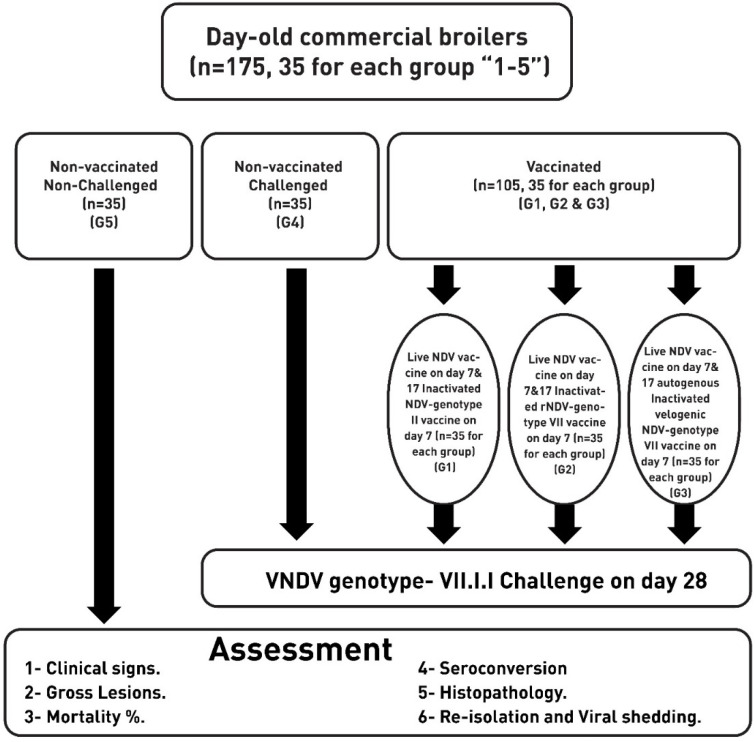
Experimental plan for assessment of inactivated NDV genotype II, inactivated rNDV genotype VII, and autogenous velogenic NDV genotype VII effectiveness against velogenic NDV-genotype VII.I.I (Accession No.:MG029120) challenge in broilers.

**Figure 2 vaccines-10-00029-f002:**
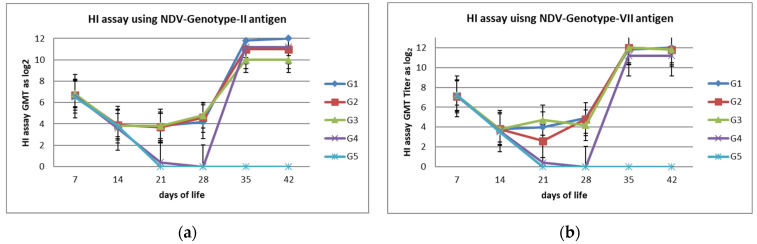
(**a**) Mean HI titer ±SD in vaccinated and non-vaccinated groups using LaSota Antigen. (**b**) Mean HI titer ±SD in vaccinated and non-vaccinated groups using NDV-genotype-VII Antigen.

**Figure 3 vaccines-10-00029-f003:**
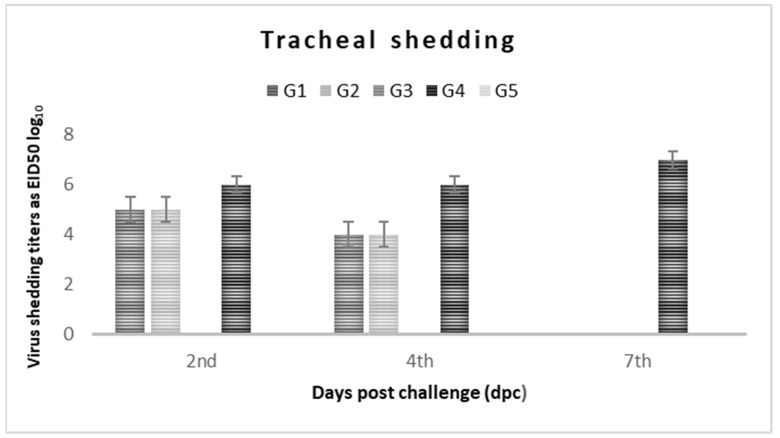
Tracheal shedding titers log_10_ EID_50_/1 mL (±SD) in vaccinated and non-vaccinated broiler chickens at dpc.

**Figure 4 vaccines-10-00029-f004:**
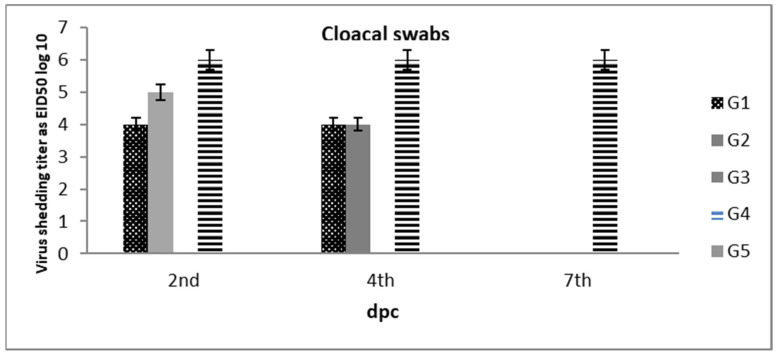
Cloacal shedding titers log_10_ EID_50_/1 mL (±SD) in vaccinated and non-vaccinated broiler chickens at dpc.

**Figure 5 vaccines-10-00029-f005:**
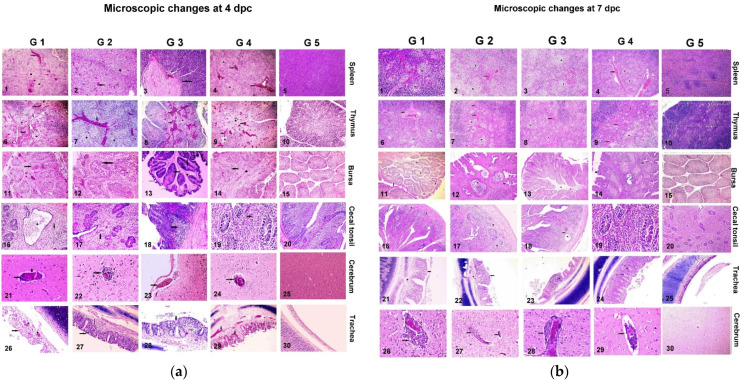
(**a**) 1: Spleen (G-1) showing severe congested blood vessels (arrow), hemorrhage, and lymphocyte depletion (50×). 2: Spleen (G-2) showing severe lymphocyte depletion (star), congestion, and hemorrhage (arrow) (50×). 3: Spleen (G-3) showing thickening of the blood vessel walls (arrow) (100×). 4: Spleen (G-4) showing congested blood vessels (arrow) and lymphocyte depletion (star) (50×). 5: Spleen (G-5) showing apparently normal structures (H&E 100×). 6: Thymus (G-1) showing severe depletion, lymphocyte necrosis (star), and congested blood vessels (arrow) (H&E 50×).7: Thymus (G-2) showing thymocyte depletion (star) with congestion of blood vessels (arrow) (H&E 50×). 8: Thymus (G-3) showing severe thymocyte depletion (star) with severe congestion and hemorrhage (arrow) (H&E 50×). 9: Thymus (G-4) showing congested blood vessels (arrow) (H&E 100×).10: Thymus (G-5) showing apparently normal structures (H&E 100×). 11: Bursa (G-1) showing severe lymphocyte depletion and degeneration (arrow) with interfollicular congestion (line) and connective tissue proliferation (star) (H&E 100×). 12: Bursa (G-2) showing severe lymphocyte depletion in cortex and medulla (arrow) with interfollicular congested blood vessels (line) (H&E 200×). 13: Bursa (G-3) showing hyperplasia of epithelium lining with cystic formation (line) and lymphocyte depletion in the medulla (arrow) with interfollicular edema (star) (H&E 100×). 14: Bursa (G-4) showing lymphocyte depletion (arrow) in medulla of follicles, congested blood vessels with microcysts formation, and interfollicular connective tissue proliferation (star) (H&E 100×). 15: Bursa (G-5) showing apparently normal structures (H&E 100×). 16: Cecal tonsil (G-1) showing cystic crypt containing necrotic debris (star) with lymphocyte depletion (arrow) (H&E 200×). 17: Cecal (G-2) showing depletion and lymphocyte necrosis (arrow) (200×). 12: Cecal tonsil (G-3) showing lymphocyte depletion (arrow) (H&E 200×). 12: Cecal tonsil (G-3) showing lymphocyte depletion (H&E arrow) (200×). 19: Cecal tonsil (G-4) showing lymphocyte depletion and necrosis (arrow) (H&E 400×). 20: Cecal tonsil (G-5) showing apparently normal structures (H&E 100×). 21: Cerebrum (G-1) showing lymphocytic perivascular cuffs (arrow) (H&E 400×). 22: Cerebrum (G-2) showing lymphocytic perivascular cuffs with perivascular edema (arrow) and neuronal degeneration (line) (400×). 23: Cerebrum (G-3) showing submeningeal congested blood vessels (arrow) (200×). 24: Cerebrum (G-4) showing perivascular cuff, large, and small blood vessels surrounded by inflammatory cells (200×). 25: Cerebrum (G-5) showing apparently normal structures (H&E 100×). 26: Trachea (G 10) showing hyperplasia of epithelium lining (arrow) with severe mucosal and submucosal congestion and edema (star) (H&E 100×). 27: Trachea (G-2) showing hyperplasia of epithelium lining (thick arrow) with mild congested blood vessels, edema activation of mucous glands, and few mononuclear cells infiltration in lamina propria (thin arrow) (200×). 28: Trachea (G-3) showing hyperplasia of epithelium lining with edema, activation of mucous glands (thick arrow), and little mononuclear cells infiltration in lamina propria (thin arrow) (H&E 200×). 29: Trachea (G-4) showing edema and severe congestion in lamina propria (arrow) (H&E 400×). 30: Trachea (G-5) showing apparently normal structures (H&E 100×). (**b**): 1: Spleen (G-1) broiler chickens 7dpc showing congested blood vessels (arrow) and lymphocyte depletion (star) (H&E 100×). 2: Spleen (G-2) broiler chickens 7dpc showing hemorrhage (arrow) with lymphocyte depletion (star) (H&E 100×). 3: Spleen (G- 3) broiler chickens showing lymphocyte depletion (star)s with focal hemorrhage (arrow) (H&E 100×). 4: Spleen (G- 4) showing congested blood vessels (arrow) (H&E 100×). 5: Spleen (G-5) showing apparently normal structures (H&E 100×).6: Thymus of (G-1) showing lymphocyte depletion (star) and congested blood (arrow) vessels (H&E 200×).7: Thymus of (G- 2) showing mild thymocyte depletion (star) with congested blood vessels (arrow) (H&E 50×). 8: Thymus of (G-3) showing thymocyte depletion (star) and congested blood vessels (arrow) (H&E 100×). 9: Thymus (G-4) showing thymocyte depletion (star) and congested blood (arrow) vessels (H&E 100×). 10: Thymus (G-5) showing apparently normal structures (H&E 100×). 11: Bursa of (G-1) showing hyperplasia of epithelium lining (thin arrow), lymphoid follicle lymphocyte depletion (stars), and epithelization (thick arrow) (H&E 200×). 12: Bursa of (G-2) showing lymphocyte depletion, epithelization (line), cyst formation (arrow), and interfollicular connective tissue proliferation (star) (H&E 100×). 13: Bursa of (G-3) showing mild lymphocyte depletion (line) with interfollicular edema (star) (H&E 100×). 14: Bursa of G 4 showing lymphocyte depletion epithelization (line), cyst formation (arrow), and proliferation of corticomedullary epithelium (star) (H&E 100×). 15: Bursa (G-5) showing apparently normal structures (H&E X100). 16: Cecal tonsils of (G-1) showing apparently normal structures (H&E 50×). 17: Cecal tonsils of (G-2) showing mild lymphocyte depletion (arrow) (H&E 100×). 18: Cecal tonsils of (G-3) showing cystic crypt in lamina propria (arrow) (H&E 100×). 19: Cecal tonsil of (G-4) showing apparently normal architectures (H&E 100×). 20: Cecal tonsil (G-5) showing apparently normal structures (H&E 100×). 21: Trachea (G-1) showing hyperplasia of epithelium lining (arrow) (H&E 200×). 22: Trachea (G-2) showing hyperplasia of epithelium lining (arrow) with edema and congestion in lamina propria (H&E 100×). 23: Trachea (G-3) showing hyperplasia of epithelium lining, mucosa thickening (line), and lymphocytic nodule (star) with edema in mucosa and submucosa (H&E 200×). 24: Trachea (G-4) 7dpc showing mucosal layer thickening(arrow), edema, congested blood vessels, and mononuclear cell infiltration with mucous gland activation (H&E 100×). 25: Trachea (G-5) showing apparently normal structures (H&E 100×). 26: Cerebrum (G-1) showing congested blood vessels and perivascular cuff (arrow) (H&E 400×). 27: Cerebrum (G-2) showing multiple perivascular cuff (arrows) (H&E 400×). 28: Cerebrum (G-3) showing multiple perivascular cuff (arrows) (H&E 400×). 29: Cerebrum (G 4) showing lymphocytic perivascular cuff (arrow) (H&E 400×). 30: Cerebrum (G-5) showing apparently normal structures (H&E 100×).

**Table 1 vaccines-10-00029-t001:** Oligonucleotide primers used for the amplification of the NDV F-protein.

Gene	Primer	Primer Sequence 5′-3′	Reference
NDV F-gene	Forward	5′-TCCGGAGGATACAAGGGTCT-3′	[31]
	Reverse	5′-AGCTGTTGCAACCCCAAG-3′	
	Probe	5′-FAM-AAGCGTTTCTGTCTCCTTCCTCCA-BHQ1-3′	

F-gene: fusion protein gene.

**Table 2 vaccines-10-00029-t002:** Clinical signs and mortality in vaccinated and non-vaccinated challenged groups on day 28 with VNDV genotype VII.I.I.

Groups	Inactivated Vaccine	Eye Lesion	Resp. Signs	Greenish Diarrhea	Nasal Discharge	Paresis	Nervous Signs	Mortality No. (%)
G1	NDV-GII	0/35 ^a^	13/35 ^b^	25/35 ^b^	3/35 ^a^	0/35	0/35	8/35 ^b^ (23%)
G2	rNDV-GVII	2/35 ^a^	10/35 ^b^	20/35 ^b^	8/35 ^a^	2/35	1/35	5/35 ^a^ (14%)
G3	Auto-NDV-GVII	0/35 ^a^	3/35 ^a^	10/35 ^a^	2/35 ^a^	0/35	0/35	1/35 ^a^ (3%)
G4	Non-Vac	15/35 ^b^	35/35 ^b^	32/35 ^b^	18/35 ^b^	2/35	1/35	32/35 ^b^ (91%)
G5	Non-Vac/Non-Chal	0/35	0/35	0/35	0/35	0/35	0/35	0/35 (0%)

G1: inactivated LaSota. G2: inactivated recombinant-LaSota GVII. G3: inactivated autogenous experimental prepared GVII. G4: Non-vaccinated challenged. G5: non-vaccinated non challenged. Non-chal: Non-challenged Auto-NDV GVII: autogenous genotype-VII NDV vaccine. Non-Vac: non-vaccinated challenged group. Resp.: respiratory. Mortality No.: number of mortalities following challenge. a,b: represent the statistical analysis significance.

**Table 3 vaccines-10-00029-t003:** Microscopic lesions score in vaccinated and non-vaccinated challenged groups on day 28 with VNDV genotype VII.I.I.

Lesion Scores at Dpc															
4 Dpc								7 Dpc							
G.	Sp	Tr	C.T	Br	Thy	Cerb	MSI	G.	Sp.	Tr.	C.T.	Br.	Thy.	Cerb.	MSI
G1	3	2	2.3	2.4	2.4	2	2.6 ± 0.78	G1	2.1	2.3	1.8^a^	2.3	2.3	2.1	1.98 ± 0.28
G2	2.3	2	2.3	2.3	1.3	2	2.0 ± 0.23	G2	2	1.5 ^a^	1.6 ^a^	1.4 ^a^	1.9	1.5	1.6 ± 0.17
G3	1.8	1.9	1.7	1.7	2	1.9	1.7 ± 0.12 ^a^	G3	1.3 ^a^	2	1.4 ^a^	1.4 ^a^	1.6 ^a^	1.3	1.25 ± 0.18 ^a^
G4	3.4	3.4	3	3.3	3.3	3.4	3.4 ± 0.5 ^b^	G4	3.5 ^b^	3.6 ^b^	3.1 ^b^	3.3 ^b^	3.3 ^b^	3.4	3.5 ± 0.2 ^b^
G5	1	1	1	1	1	1	1 ± 0.0	G5	1	1	1	1	1	1	1 ± 0.0

G.: group; Sp = spleen. Tr: trachea. C.T: cecal tonsils. Br: brain. Thy: thymus. Creb: cerebellum. MSI: the formula of the mean severity index is given by the function sum of mean lesions score. a,b: represent the statistical analysis significance.

**Table 4 vaccines-10-00029-t004:** Mean HI titers, mortality percentages, and viral shedding in vaccinated and non-vaccinated challenged groups on day 28 with VNDV genotype VII.I.I.

G	HI ± SD	Viral Shedding ^1^											
		2nd Dpc				4th Dpc				7th Dpc			
		Tr		Cl		Tr		Cl		Tr		Cl	
		Pos ^2^/Ex No.	Mean ± SD	No/Ex	Mean ± SD	No/Ex	Mean ± SD	No/Ex	Mean+ SD	No/Ex	Mean ± SD	No/Ex	Mean ± SD
**G1**	4.2 ± 0.45	4/10 (40%)	6.3 × 105 ± 1.2 ^a^	4/10 (40%)	4 × 105 ± 1.4 ^a^	4/10 (40%)	9.4 × 104 ± 2.3 ^a^	4/10 (40%)	1.6 × 104 ± 1.32 ^a^	0/10 (0%)	0	0/10 (0%)	0
**G2**	4.2 ± 0.55	2/10 (20%)	2.3 × 105 ± 0.0 ^a^	2/10 (20%)	5.5 × 104 ± 0.0 ^a^	2/10 (20%)	1.4 × 104 ± 0.0 ^a^	4/10 (40%)	4 × 104 ± 1.3 ^a^	0/10 (0%)	0	0/10 (0%)	0
**G3**	4.8 ± 0.67	2/10 (20%)	2.1 × 105 ± 0.0 ^a^	2/10 (20%)	4.5 × 104 ± 0.0 ^a^	2/10 (20%)	1.1 × 104± 0.0 ^a^	2/10 (20%)	3 × 104 + 0.0 ^a^	0/10 (0%)	0	0/10 (0%)	0
**G4**	nd	6/10 (60%)	1.5 × 106 ± 1.4 ^b^	6/10 (60%)	1.3 × 106 ± 0.8 ^b^	10/10 (100%)	6.2 × 106 ± 1.76 ^b^	10/10 (100%)	4.6 × 106 ± 1.31 ^b^	10/10 (100%)	1.7 × 107 ± 0.87	10/10 (100%)	2.6 × 106 ± 0.63
**G5**	nd	0/10 (0%)	Nd	nd	nd	nd	nd	nd	nd	nd	nd	nd	nd

G: Group (*n* = 10). HI: mean HI titer at the day challenge ± SD CL: cloacal swab; dpc: day post challenge; NDV: Newcastle disease virus; RT-PCR: reverse transcriptase polymerase chain reaction; TR: tracheal swab. nd non-detectable level. G1: genotype II live/genotype II inactivated and challenged; G2: genotype II live/genotype VII inactivated and challenged; G3: genotype II live/velogenic G VII inactivated and challenged; G4: unvaccinated and challenged. ^1^ Virus shedding was tested by RT-PCR, ^2^ Pos/Ex no. = positive number/examined number. a,b: represent the statistical analysis significance.

## Data Availability

All data included in the manuscript.

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
