# Peer review of "Efficacy of the Newcastle Disease Virus Genotype VII.1.1-Matched Vaccines in Commercial Broilers"

_vaccines, 2021, doi:10.3390/vaccines10010029_

Round 1

Reviewer 1 Report

This manuscript examined various NDV vaccines for Genotype VII.1.1 in commercial broilers. There are items that need to be addressed before a decision about publishing can be made.

Line 22, 23 the word "tooK' is inappropriate term for inoculation of the groups.

Line 25 grammar issues with the sentence.

Line27 plus many additional lines replace "histopathological changes" with "microscopic changes".

Line 35. Delete "So," and capitalize " updating..."

Line 50,51- remove parentheses from the list of proteins.

Line 51- remove "are" from "proteins are forming"

Line 67- start a new paragraph with Antigenic similarity....

Line 68 replace "So," with "Thus,"

line 70 remove space between "diversity of" and "the different..."

Line 77 Insert "to" between  "according" and "the recent.."

LIne 104 Change "in broilers chicks" to In broiler chicks, NDV is associated"

LInes107-108- Disease names need standardization--capitalize each word or not.

Line 109  Follicular degeneration of what? Bursa or ovary?

Material and Methods-- Do the commercial vaccines need the Trademark symbol or not

Line 129 insert "was" between "virus" and "the"

Line 143 change "described on OIE manual " to "described in the OIE manual"

Line 150 change "flock which vaccinated" to "flock that was vaccinated"

There is no true negative control group that was not vaccinate nor challenged in your experimental design.

Line 172-- When was serology collected? not clear at all

Line 183 "severity"  severity based on what?

Line 187-- How were the birds euthanized-- this may affect your lesion scoring.

Line 220  What kind of eye lesions-- conjunctivitis, swollen sinuses, ulcerations of the cornea?

Line 243-- which birds were the serological responses examined?

Line 260-264  Tissue names should not be capitalized when in a list.

Line 271 "sever" is spelt wrong

Figure  1 -- Text prints out 2 times on my copy for some reason

Figure 2a/b-- Axis need labels in both figures

Figure 5a--many of the photomicrographs are out of focus and you can't see the arrow or stars etc that are used as labels in photos 9-12

Line 333 Are you saying there is smooth muscle hyperplasia due to the infection/vaccination?

Figure 5b. Hard to see labels on photos. Staining is off on photos 4, 5 and 6. Nervous tissue is pink not blue.  Why isn't photo 12 with the rest of the thymic photos in the plate?

Figure 5C. Non-supperative is not a term used in avian histopathology as birds do not form pus so call it lymphocytic. Photo 9 appears out of focus--can't tell apart the heterophils or eosinophils.  With no negative control, how do you know that the congestion is real vs method of euthanasia?

Figure 5D-- Hard to see the labels again-- remove non-suppurative terminology.

Line 406-413 is a long run-on sentence.

Line 435--are the lesions more or less severe-- incomplete sentence

Line 446-450 is a long run-on sentence

Line 479-- Cerebellum demonstrated... All the photomicrographs appear to be of the cerebrum, not cerebellum

Line 483  "significantly lower"  the score is actually higher than G-2 and G-3.

Line 497. Table should be capitalized

References-- Follow the journal format for journal names. You have some that are spelled out and others with abbreviations of the name.

Line 671 is an incomplete citation-- no journal name given at all or date of publication

Author Response

Response for Reviewer-1

This manuscript examined various NDV vaccines for Genotype VII.1.1 in commercial broilers. There are items that need to be addressed before a decision about publishing can be made.

Q: Line 22, 23 the word "tooK' is inappropriate term for inoculation of the groups.

Response: changed to received

Line 25 grammar issues with the sentence.

Response: corrected

Line27 plus many additional lines replace "histopathological changes" with "microscopic changes".

Response: changed

Line 35. Delete "So," and capitalize " updating..."

Response: deleted and capitalized

Line 50,51- remove parentheses from the list of proteins.

Response: removed

Line 51- remove "are" from "proteins are forming"

Response: Removed

Line 67- start a new paragraph with Antigenic similarity....

Response: Done

Line 68 replace "So," with "Thus,"

Response: Done

line 70 remove space between "diversity of" and "the different..."

Response: removed

Line 77 Insert "to" between  "according" and "the recent.."

Response: Done

LIne 104 Change "in broilers chicks" to In broiler chicks, NDV is associated"

Response: changed  

LInes107-108- Disease names need standardization--capitalize each word or not.

Response: standardized and capitalize each word

Line 109  Follicular degeneration of what? Bursa or ovary?

Response: Added to line 111 “ovary and oviducts)

Material and Methods-- Do the commercial vaccines need the Trademark symbol or not

Response: Actually, we don’t have full data about registering the trademark in Egypt or in the of country origin, the laws in Egypt may allow selling registered vaccine in the Governmental veterinary authority without trademark registrations, so we did not mention it

Line 129 insert "was" between "virus" and "the"

Response: inserted

Line 143 change "described on OIE manual " to "described in the OIE manual"

Response: Changed

Line 150 change "flock which vaccinated" to "flock that was vaccinated"

Response: Changed

There is no true negative control group that was not vaccinate nor challenged in your experimental design.

Response: it was miss wrote, we added the G5 (non-vaccinated non challenged group to the design and the histopathology photos)

Line 172-- When was serology collected? not clear at all

Response:  added on line 177-178 “Blood samples were collected on 7-, 14-, 21-, 28-, 35- and 42-day of life and serum separated and stored on -20c until analysis”

Line 183 "severity"  severity based on what?

A: we added a reference to our scoring system

Line 187-- How were the birds euthanized-- this may affect your lesion scoring.

Response:  using the cervical dislocation; we included the non-vaccinated non-challenged groups (G5) to the histopathology and included in its results to eliminate any effect of the experimental normal operation like rearing conditions, euthanasia process… and to ensure proper standerization to the proposed scoring system for lesions.

Line 220 What kind of eye lesions-- conjunctivitis, swollen sinuses, ulcerations of the cornea?

Response:  added on line 223-224 “mild or moderate respiratory signs (nasal discharge, respiratory rules, and 71.5% (25/35) showed greenish diarrhea, but no eye lesions (lacrimation, slight occlusion to eye) and most recovered soon, with 73% (26/35) clinical protection within 10-dpc”

Line 243-- which birds were the serological responses examined?

Response:  Added on line 249-250 “blood samples collected from all groups on weekly interval starting from 7-day of life till 42-day of life”

Line 260-264 Tissue names should not be capitalized when in a list.

Response:  changed “various organs including trachea, lung, bursa, liver and spleen”

Line 271 "sever" is spelt wrong

Response:  corrected “severe in non-vaccinated group.”

Figure 1 -- Text prints out 2 times on my copy for some reason

Response:  only inserted once

Figure 2a/b-- Axis need labels in both figures

Response:  Added to the Figures 2a/b

Figure 5a--many of the photomicrographs are out of focus and you can't see the arrow or stars etc that are used as labels in photos 9-12

Response: the figure 5a updated and corrected and update the photos

Line 333 Are you saying there is smooth muscle hyperplasia due to the infection/vaccination?

Response:  yes, may be the thickening of blood vessels is due to hyperplasia of VSMC

Figure 5b. Hard to see labels on photos. Staining is off on photos 4, 5 and 6. Nervous tissue is pink not blue.  Why isn't photo 12 with the rest of the thymic photos in the plate?

Response:  the figure 5b updated and corrected and update the photos

Figure 5C. Non-supperative is not a term used in avian histopathology as birds do not form pus so call it lymphocytic. Photo 9 appears out of focus--can't tell apart the heterophils or eosinophils.  With no negative control, how do you know that the congestion is real vs method of euthanasia?

Response: the figure 5c updated and the non-supperative term removed, the results compared with control tissue.

Figure 5D-- Hard to see the labels again-- remove non-suppurative terminology.

Line 406-413 is a long run-on sentence.

Response: Rewrote

Line 435--are the lesions more or less severe-- incomplete sentence

A: changed line 439-444 “The degree of matching between the applied vaccines and challenged virus decreases from autogenous NDV vaccines to rNDV-GVII vaccines then to G-II vaccines, and thus, may be associated with the variation in the severity of gross lesions as birds in G-1 showed more severe gross lesion followed by G2 then G3 which confirmed the proportional relationship between the close degree of matching (between vaccine and challenge virus) and the protection levels.”

Line 446-450 is a long run-on sentence

Response: Rewrote

Line 479-- Cerebellum demonstrated... All the photomicrographs appear to be of the cerebrum, not cerebellum

Response: Corrected and changed to Cerebrum all over the manuscript

Line 483  "significantly lower"  the score is actually higher than G-2 and G-3.

Line 497. Table should be capitalized

Response: capitalized “showed (tableTable-4 and Figure-3 and -4)”

References-- Follow the journal format for journal names. You have some that are spelled out and others with abbreviations of the name.

Response: updated all reference with endnote MDPI reference style

Line 671 is an incomplete citation-- no journal name given at all or date of publication

Response: updated “Van Boven, M.; Bouma, A.; Fabri, T.H.; Katsma, E.; Hartog, L.; Koch, G. Herd immunity to Newcastle disease virus in poultry by vaccination. Avian pathology 2008, 37, 1-5, doi:https://doi.org/10.1080/03079450701772391.”

Reviewer 2 Report

Although interesting in general, there are some aspects that should be revised in order to improve the clarity of this manuscript.

 Line 7 - Is "e-mail@e-mail.com" a real email address? It looks very unprofessional

Line 170 – the term “dpc” should be abbreviated as used for the first time in manuscript (except the abstract)

Line 180 - “2.5 Histopathologic examination.” or (line 250) “3.3 Histopathological examination”? I think the correct is “Histopathological examination”.

Line 181 – why brain is written in capital letters?

Line 182 – how long the tissue samples were fixed with the buffered formalin ?

Line 183 – the authors used a semi-quantitative scale for severity of histopathological changes assessment but they did not present what were the criteria ?

Line 239, 339, 479 – the authors stated that they observed cell death (necrosis). How can they be sure that the at least some observed changes were not apoptotic in nature? Did they use any method that can distinguish apoptosis from necrosis (cells morphology, surface markers or flow fluorocytometry)?

Lines 248-350 - from an anatomical point of view, cerebrum is not synonymous with the brain. The cerebrum, or telencephalon, is the large upper part of the brain.

Line 303 – some organs are written in capital letters some not. Please unify.

Line 359 – why “changes” are written bold ?

Author Response

Response for Reviewer-2

Although interesting in general, there are some aspects that should be revised in order to improve the clarity of this manuscript.

Line 7 - Is "e-mail@e-mail.com" a real email address? It looks very unprofessional

Response:  removed

Line 170 – the term “dpc” should be abbreviated as used for the first time in manuscript (except the abstract)

Response:  abbreviated on line 30 “against mortality at 10–days post-challenge (dpc).”

Line 180 - “2.5 Histopathologic examination.” or (line 250) “3.3 Histopathological examination”? I think the correct is “Histopathological examination”.

Response:  corrected to Histopathological

Line 181 – why brain is written in capital letters?

Response:  changed to small letter

Line 182 – how long the tissue samples were fixed with the buffered formalin ?

Response:  added to line 187 “ Tissue samples (trachea, cecal tonsils, bBrain, thymus, and spleen) were fixed in 10% buffered formalin for about 15 to 20 days, “

Line 183 – the authors used a semi-quantitative scale for severity of histopathological changes assessment but they did not present what were the criteria ?

Response:  Scoring reference added

Line 239, 339, 479 – the authors stated that they observed cell death (necrosis). How can they be sure that the at least some observed changes were not apoptotic in nature? Did they use any method that can distinguish apoptosis from necrosis (cells morphology, surface markers or flow fluorocytometry)?

Response:  thanks for these great comments, it confirmed by cells morphology the necrosis which was described was not apoptosis because there are any apoptotic bodies which usually appear as vesicle containing necrotic cell debris which can be seen by the assistance of light microscope.

It added to line 253-257 “severe focal necrosis in the spleens of G-1, G-2 and G-3 vs. very severe necrosis in G-4, respectively, the necrosis confirmed and differentiated from the apoptotic examined the slide field by light microscope and there are no apoptotic bodies which usually appear as vesicle containing necrotic cell debris thus confirmed the lesions is necrosis not apoptotic lesions.”

Lines 248-350 - from an anatomical point of view, cerebrum is not synonymous with the brain. The cerebrum, or telencephalon, is the large upper part of the brain.

Response:  updated and described the microscopic lesions of cerebrum

Line 303 – some organs are written in capital letters some not. Please unify.

Response:  changed to small

Line 359 – why “changes” are written bold ?

Response:  corrected

Round 2

Reviewer 1 Report

The revised manuscript is better. There are just some minor changes suggested.

Line 25. Insert "were" between groups and challenge-- added to "challenged"

Line 33' replace "close" with "closely"

Line 151- remove the - between five and day old otherwise is reads like 170, 5 day old chicks vs 175 day-old chicks.

Line 186- replace "applied" with "was performed"

Line 227. replace "rules" with "rales"

Line 259 insert "with" between "HI test," and "blood samples..."

Line 273. Replace Histopathological with Microscopic

Line 294 grammar.. "whiles" should be "while", "route" should be "routes"

Line 304/305 same as above

Line 321. Replace "Histopathological" with "Microscopic"

Figure 5a and 5b-- typically magnification is expressed 50X not X50 but will defer to journal format

Line 433 "rresults" is spelt wrong

Line 444,. Remove "took"--not correct term

Citation 49. Incomplete citation--no journal name/volume etc or DOI information.

Author Response

Dear Reviewer

Greeting

We are truly grateful to yours and other reviewers’ critical comments and thoughtful suggestions on our manuscript (Efficacy of the Newcastle disease virus genotype VII.1.1 matched vaccines in commercial broilers). Based on these comments/suggestions, we have made careful modifications on the original manuscript. All changes made to the text are in red color.

Yours

Authors

The point-to-point reviewers’ comments

Reviewers’ comments

Reviewer-2

The revised manuscript is better. There are just some minor changes suggested.

Response: Highly appreciated the reviewer comments and advises

Line 25. Insert "were" between groups and challenge-- added to "challenged"

Response: Inserted

Line 33' replace "close" with "closely"

Response: changed

Line 151- remove the - between five and day old otherwise is reads like 170, 5 day old chicks vs 175 day-old chicks.

Response: removed

Line 186- replace "applied" with "was performed"

Response: replaced

Line 227. replace "rules" with "rales"

Response: replaced

Line 259 insert "with" between "HI test," and "blood samples..."

Response: inserted

Line 273. Replace Histopathological with Microscopic

Response: replaced

Line 294 grammar.. "whiles" should be "while", "route" should be "routes"

Response: corrected

Line 304/305 same as above

Response: deleted

Line 321. Replace "Histopathological" with "Microscopic"

Response: Replaced

Figure 5a and 5b-- typically magnification is expressed 50X not X50 but will defer to journal format

Response: changed to 50X, 100X, 200X

Line 433 "rresults" is spelt wrong

Response: corrected

Line 444,. Remove "took"--not correct term

Response: changed to received

Citation 49. Incomplete citation--no journal name/volume etc or DOI information

Response: added journal name and volume No.